# Lateral orbitofrontal cortex promotes trial-by-trial learning of risky, but not spatial, biases

Christine M Constantinople[1†*], Alex T Piet[1‡], Peter Bibawi[1], Athena Akrami[1,2,3§], Charles Kopec[1,2], Carlos D Brody[1,2,3]

[1]Princeton Neuroscience Institute, Princeton University, Princeton, United States; [2]Department of Molecular Biology, Princeton University, Princeton, United States; [3]Howard Hughes Medical Institute, Princeton University, Princeton, United States

**Abstract** Individual choices are not made in isolation but are embedded in a series of past experiences, decisions, and outcomes. The effects of past experiences on choices, often called sequential biases, are ubiquitous in perceptual and value-based decision-making, but their neural substrates are unclear. We trained rats to choose between cued guaranteed and probabilistic rewards in a task in which outcomes on each trial were independent. Behavioral variability often reflected sequential effects, including increased willingness to take risks following risky wins, and spatial 'win-stay/lose-shift' biases. Recordings from lateral orbitofrontal cortex (lOFC) revealed encoding of reward history and receipt, and optogenetic inhibition of lOFC eliminated rats' increased preference for risk following risky wins, but spared other sequential effects. Our data show that different sequential biases are neurally dissociable, and the lOFC's role in adaptive behavior promotes learning of more abstract biases (here, biases for the risky option), but not spatial ones.
DOI: https://doi.org/10.7554/eLife.49744.001

*For correspondence:
constantinople@nyu.edu

Present address: †Center for Neural Science, New York University, New York, United States; ‡Allen Institute for Brain Science, Seattle, United States; §Sainsbury Wellcome Centre, London, United Kingdom

Competing interests: The authors declare that no competing interests exist.

## Introduction

Sequential biases permeate human decision-making, and while using past experiences to guide decision-making can be useful in dynamic environments, it can cause us to make suboptimal decisions if the past is not informative (but see *Yu and Cohen, 2008*). A variety of biases have been described in two-alternative forced choice tasks in humans and animal models, including repetition of successful choices ('win-stay'), switching after unsuccessful choices ('lose-switch'), and biases due to previous sensory experiences (*Abrahamyan et al., 2016*; *Akrami et al., 2018*; *Busse et al., 2011*; *Hollingworth, 1910*; *Kagel et al., 1995*; *Scott et al., 2015*; *Visscher et al., 2009*). Moreover, depending on task design, these biases can be expressed in different ways: if choice options are fixed in space, as is often the case in studies of rodent behavior, sequential biases will appear spatial and action-dependent. Alternatively, if the choice options are not fixed in space, sequential biases may be expressed in non-spatial, task-dependent coordinates. It is unclear whether different sequential biases, or biases expressed in different coordinate reference frames, rely on shared or distinct neural circuits and mechanisms.

We trained rats to choose between explicitly cued guaranteed or probabilistic (i.e., risky) rewards (*Constantinople et al., 2019*). Guaranteed and risky rewards were randomly assigned to different locations on each trial, disambiguating biases expressed in spatial coordinates and those expressed in more abstract coordinates for the task (biases for risky or safe options). We found that the lateral orbitofrontal cortex (lOFC) was required for rats' expression of abstract, but not spatial, sequential

biases. We interpret these data as consistent with proposals that OFC represents the animal's current location within an abstract cognitive map of the task (*Wilson et al., 2014*).

The OFC is not a monolithic structure: in rats, subdivisions (e.g., ventral orbital area, lateral orbital area, agranular insula) are characterized by distinct efferent and afferent projections and, presumably as a consequence, there is growing evidence that these subdivisions make distinct functional contributions to behavior (*Dalton et al., 2016*; *Groenewegen, 1988*; *Hervig et al., 2019*; *Izquierdo, 2017*; *Murphy and Deutch, 2018*; *Ray and Price, 1992*). Based on connectivity, the rat lOFC (including lateral orbital and agranular insular areas) is thought to be homologous to the central-lateral OFC of monkeys (*Heilbronner et al., 2016*; *Izquierdo, 2017*; *Stalnaker et al., 2015*), although differences have been observed in these areas across species, for example in neural dynamics following reversal learning (*Morrison et al., 2011*; *Schoenbaum et al., 1999*).

The rodent lOFC and primate central-lateral OFC have been shown to play critical roles in adapting behavior to dynamic task contingencies, especially when those contingencies are partially observable (*Rushworth et al., 2011*; *Stalnaker et al., 2014*; *Wallis, 2007*; *Wilson et al., 2014*). Lesion experiments have implicated the lOFC in tracking rewards and value, for example in extinction, devaluation, and reversal learning paradigms, in which reward contingencies were explicitly manipulated by the experimenter (*Gallagher et al., 1999*; *Izquierdo et al., 2004*; *Pickens et al., 2003*). A related body of work implicates the lOFC in evaluative processes, including comparing current choices to previous outcomes (*Kennerley et al., 2011*), and regret (*Steiner and Redish, 2014*; *Steiner and Redish, 2012*).

Most of the studies demonstrating that OFC promotes behavioral flexibility have used tasks in which trial-by-trial learning improves behavioral performance (e.g., reversal learning). We hypothesized that OFC's role in behavioral flexibility might drive sequential biases even when they are deleterious. We developed a task in which sequential biases were maladaptive: trials were independent and reward contingencies were stable over months of training (*Constantinople et al., 2019*). Rats exhibited several dissociable biases reflecting trial and reward history, and we identified lOFC as critical for one bias in particular: an increased willingness to take risk following risky wins.

## Results

### Rats choosing between guaranteed and probabilistic rewards show a risky 'win-stay' bias

We developed a task in which rats chose between explicitly cued guaranteed and probabilistic rewards (*Constantinople et al., 2019*). Animals initiated a trial by nose-poking in the center of three ports. Auditory clicks were presented from left and right speakers, and the click rate (6–48 Hz) conveyed the volume of water reward baited at each of the two side ports. Simultaneously, light flashes were presented from side ports, and the number of flashes (0–10) conveyed the probability of receiving water reward at each port (*Figure 1A*). One port offered a guaranteed or 'safe' reward (p=1), and the other offered a probabilistic or 'risky' reward (p≤1). Rewards were delivered 100 ms after rats entered the side port. The location of the safe and risky ports varied randomly on each trial. Four possible water volumes were offered (6, 12, 24, 48 μL), and risky reward probabilities ranged from 0 to 1, in increments of 0.1 (*Figure 1B,C*). The trials were self-paced, and following a choice, rats were free to initiate the next trial within 100–200 ms, although they typically took longer (*Figure 1—figure supplement 1*). However, if animals terminated the trial prematurely by breaking center fixation, they were penalized with a time-out penalty (1.5–8 s, adjusted across rats as needed).

The 'cue period' is the period when the rat is center poking, and flashes and clicks are presented. The 'choice reporting period' begins when the rat exits the center poke and is free to report his choice by poking in one of the side ports.

To determine when rats were sufficiently trained to understand the meaning of the cues in the task, we evaluated the 'efficiency' of their choices, by comparing their mean expected value (probability x reward) per trial relative to an agent that always chose the option with greater expected value ('ideal performance') and one that chose randomly ('random'; *Figure 1D*; see also *Rustichini et al., 2017*). While variable, most rats learned the meaning of the cues within 1–2 months of training in the final training stage (*Figure 1D*). Well-trained rats (n = 36 animals) performed, on

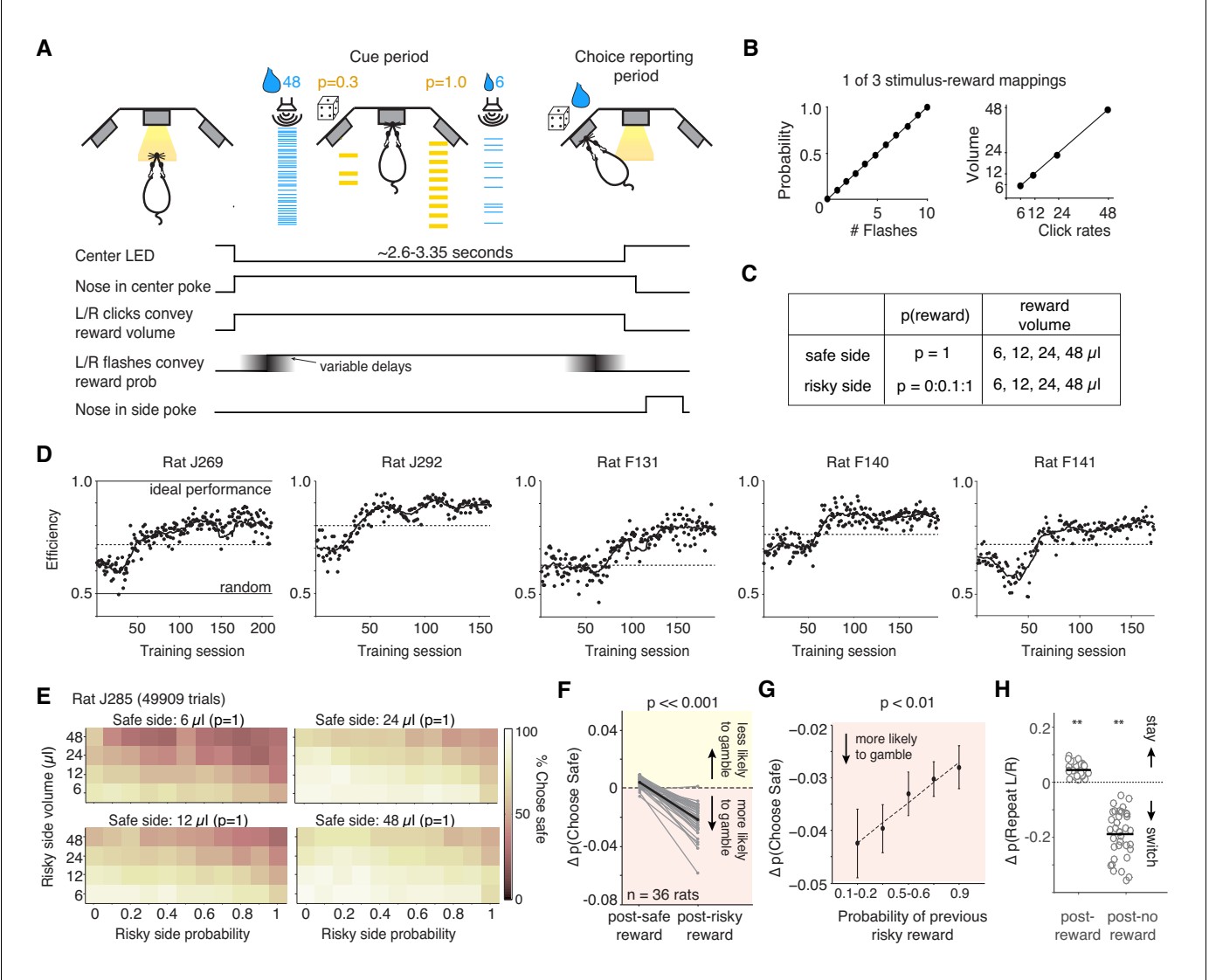

**Figure 1.** Behavioral task: Rats performing the task exhibit stable performance over months, but also trial-by-trial learning dynamics. (**A**) Example trial: rat initiates a trial by nose-poking and fixating in center. On each side, light flashes and click rates convey reward probability and water volume, respectively. One side (here, the right port) offers guaranteed reward ('safe'); safe and risky sides vary randomly over trials. (**B**) Relationship between flashes and probability, and click rates and reward volumes (6, 12, 24, or 48 μL) in one version of the task. Risky side could have rewarded probability between 0–1 (increments of 0.1). (**C**) Offered reward volumes and probabilities. (**D**) Behavioral performance in units of 'efficiency' for five representative rats in the final training stage (Materials and methods). We compared the average expected value (reward x probability) per trial the rat received compared to an agent choosing randomly, or one that always chose the option with the greater expected value ('ideal performance'). The dashed line is criterion performance for each rat (see 'Materials and methods'). (**E**) Percent of trials one rat chose the safe option for each of the four safe volumes. Axes show probability and volume of risky alternatives. (**F**) Difference in probability of choosing the safe option following guaranteed rewards and risky rewards (relative to the mean probability of choosing safe) for all rats (black is mean). Rats were more likely to gamble following risky rewards (p=8.35e-16, paired t-test). (**G**) The magnitude of the risky win-stay bias exhibits graded dependence on the reward probability of the gamble (mean across rats). p=0.0035 of slope parameter of least-squares regression line (dashed line). The riskier the gamble that won, the more likely that rats will choose to gamble again. See also **Figure 1—figure supplement 1**. (**H**) Change in the probability of repeating left or right choices following rewarded or unrewarded trials. Asterisks indicate that rats' 'win-stay' biases were significantly different from zero (p=2.06e-13, paired t-test), as were their 'lose-switch' biases (p=2.65e-15).

DOI: https://doi.org/10.7554/eLife.49744.002

The following figure supplement is available for figure 1:

**Figure supplement 1.** Supplemental behavioral analyses.

DOI: https://doi.org/10.7554/eLife.49744.003

average, 368 trials per day (± 28 trials, s.e.m.; *Figure 1—figure supplement 1B*). They tended to choose the option with the higher expected value; on trials when both sides offered certain rewards, rats chose the larger reward, and when one side offered no reward (p=0), they chose the alternative (*Figure 1D,E*, *Figure 1—figure supplement 1C*; *Constantinople et al., 2019*).

Rats also exhibited sequential biases observed in primates: if they chose the risky option and were rewarded on the previous trial, they were more likely to gamble and choose the risky side again (*Figure 1F*; *Blanchard et al., 2014*; *Hayden et al., 2008*; *Neiman and Loewenstein, 2011*). The change in probability of choosing the risky side following a risky win was significantly different from zero (p=1.29e-13, paired t-test across rats). This bias was not due to rats 'un-learning' the meaning of the flashes: there was no change in performance on trials where both sides offered the same reward volume, in which case the better option is the guaranteed reward, indicated by the flashes (p=0.397, paired t-test across rats). The magnitude of the risky win-stay bias depended on the risky reward probability (*Figure 1G*). We emphasize that in this task, the belief that a risky 'win' increases the probability of future wins is a fallacy: outcomes are independent on each trial, by design. This result did not reflect biased estimates of conditional probabilities, often observed in finite sequential data (*Figure 1—figure supplement 1D,E*). There was no change in rats' willingness to choose the gamble following a risky loss, indicating that rats were not simply going through phases of preference for risky or safe options (p=0.79 paired t-test; *Figure 1—figure supplement 1F*). Similarly, there was no systematic change in rats' risk preferences over the course of the training session: we examined trials where the guaranteed and risky reward had the same expected value. Rats' choices on these trials indicate their risk attitudes, with risk averse rats preferring the guaranteed reward and risk seeking rats preferring the gamble (*Constantinople et al., 2019*). There was no significant change in the probability of choosing the safe option on these trials, comparing the first and second half of trials in each session (p=0.15, paired t-test), again indicating that the risky win-stay bias was not due to slow fluctuations in rats' risk preferences. Finally, because the risky and safe ports switch on each trial, this risky win-stay bias was independent of a spatial win-stay/lose-switch bias for the left or right ports (*Figure 1H*), or repetitive behavioral patterns such as perseveration (*Miller et al., 2017*). It is notable, however, that the magnitude of rats' spatial win-stay/lose-switch biases was much larger than the magnitude of their risky win-stay biases (*Figure 1F,H*).

## lOFC represents reward history during the cue period

We performed tetrode recordings in lOFC while rats performed the task. Many lOFC neurons exhibited transient responses at trial initiation, the magnitude of which reflected whether the previous trial was rewarded (*Figure 2A*). To quantify this, we measured the mean discriminability (*d'*) of firing rates at each time point comparing trials following rewarded and unrewarded choices (*Figure 2B*), analyzing cells with significantly different spike counts on these trials (n = 512 of 1459 units; p<0.05, unpaired t-test). lOFC neurons reflected reward history most strongly at trial initiation, when the animal has no information yet about the prospects on the current trial (*Figure 2B*; *Figure 2—figure supplement 1*; *Nogueira et al., 2017*). A significantly greater fraction of neurons exhibited higher firing rates following unrewarded trials, compared to rewarded trials, consistent with adaptation effects (*Figure 2C*; *Conen and Padoa-Schioppa, 2019*; *Kennerley et al., 2011*; *Padoa-Schioppa, 2009*; *Saez et al., 2017*; *Zimmermann et al., 2018*).

Our data show that lOFC neurons encode information about reward history (Figure 2A,B), which might be expected from neurons mediating trial history biases. Given that behavior likely requires the activity of populations of neurons, we next sought an unsupervised description of simultaneously recorded neurons. We employed an extension of principal components analysis, tensor components analysis (*Williams et al., 2018*); TCA, also known as CANDECOMP/PARAFAC tensor decomposition (*Figure 2D*). This method extracts features of three aspects of neural data: (1) neuron factors, which weight each neuron's activity; (2) temporal factors, which capture time-varying dynamics within a trial; and (3) trial factors, which capture dynamics across trials. TCA/PARAFAC provides a low dimensional description of neural dynamics both *within* and *across* trials, allowing for simple descriptions of complex, multi-neuron responses across multiple timescales. TCA provides a key advantage over more common dimensionality reduction techniques like PCA or Factor Analysis which require averaging over trials; TCA allows us to independently quantify trial to trial fluctuations in neural activity. TCA decomposes a 3rd order data tensor $X_{n,t,k}$ (with n neurons over k trials of length t) by a sum of

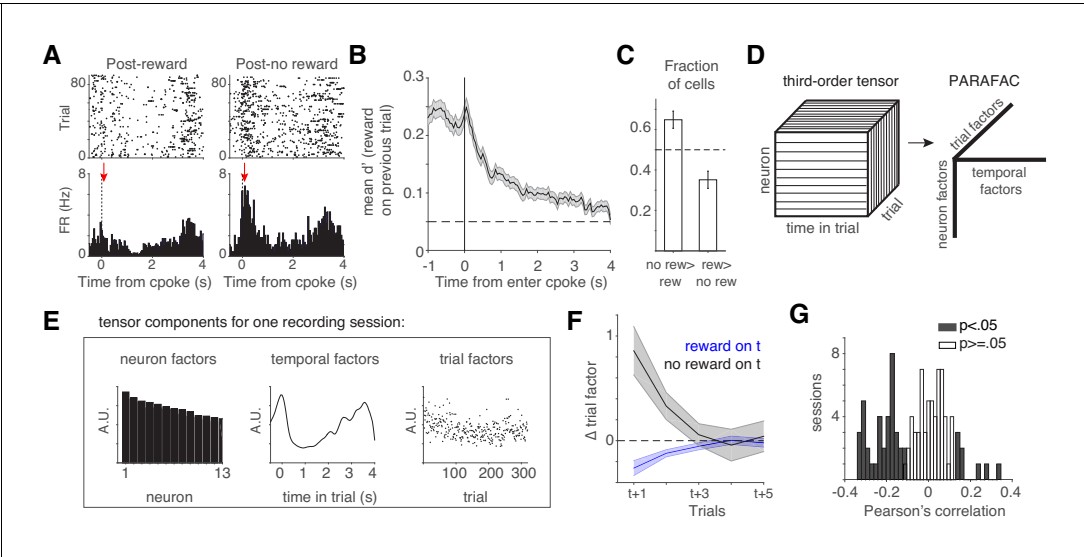

**Figure 2.** lOFC encodes reward history during the cue period. (**A**) lOFC neuron with activity aligned to trial initiation. This neuron's firing rate reflected whether the previous trial was rewarded. (**B**) Mean encoding of reward history (discriminability or d') across lOFC neurons that exhibited significantly different spike counts based on reward history. Mean ± s.e.m. See also *Figure 2—figure supplement 1*. (**C**) Fraction of neurons with significantly different spike counts based on reward history, with more spikes following unrewarded (no rew >rew) or rewarded (rew >no rew) trials. (**D**) Schematic of analysis (TCA/PARAFAC) used to discover low dimensional descriptions of trial-by-trial population dynamics. See also *Figure 2—figure supplement 2*. (**E**) Result of TCA/PARAFAC from one recording session. Y-axis is in arbitrary units (A.U.; see Materials and methods). (**F**) Mean (± s.d.) shuffle-corrected reward (blue) and no-reward (black) triggered averages of trial factors across all sessions (see Materials and methods). (**G**) Correlation between trial factors and reward history for each session. Gray bars indicate significance.

DOI: https://doi.org/10.7554/eLife.49744.004

The following figure supplements are available for figure 2:

**Figure supplement 1.** Method for identifying putatively identical waveforms over days.
DOI: https://doi.org/10.7554/eLife.49744.005

**Figure supplement 2.** TCA/PARAFAC tensor decomposition applied to neural data.
DOI: https://doi.org/10.7554/eLife.49744.006

rank 1 factors $\sum_{r=1}^{R} w_r b_r a_r$ . Here, for each rank r, w is a vector of neuron factors, b is a vector of temporal (time within trial) factors, and a is a vector of across trial factors. *Figure 2E* shows the neuron, temporal, and trial factors for one recording session (see Materials and methods, *Figure 2—figure supplement 2*).

Trial factors were significantly modulated by reward history two trials in the past, on average (*Figure 2F*). This modulation required the neurons whose firing rates reflected reward history (*Figure 2—figure supplement 2E–F*). On 43% of recording sessions (45/105 sessions), there was a significant correlation between rats' reward history and the trial factors, and these correlations were typically negative (*Figure 2G*). The trial factor can be thought of as a gain factor applied to the population response (*Figure 2—figure supplement 2D*); a negative correlation suggests that when rats received reward, neural activity in lOFC generally decreased on the subsequent trial, consistent with several reports of reward adaptation in OFC (*Conen and Padoa-Schioppa, 2019*; *Kennerley et al., 2011*; *Padoa-Schioppa, 2009*; *Saez et al., 2017*; *Zimmermann et al., 2018*). We note that there was no systematic relationship between the correlation of rats' reward history and the trial factors, and the number of simultaneously recorded neurons in each session (*Figure 2—figure supplement 2H,I*).

Given the strong encoding of aggregated reward history, we hypothesized that disrupting lOFC dynamics during the cue period would disrupt trial-by-trial learning. Optogenetic inhibition during the cue period, however, did not affect spatial win-stay or lose-switch biases (*Figure 3A–D*; *Figure 3—figure supplement 1*). Photoinhibition also did not affect the risky win-stay bias, in wild-type

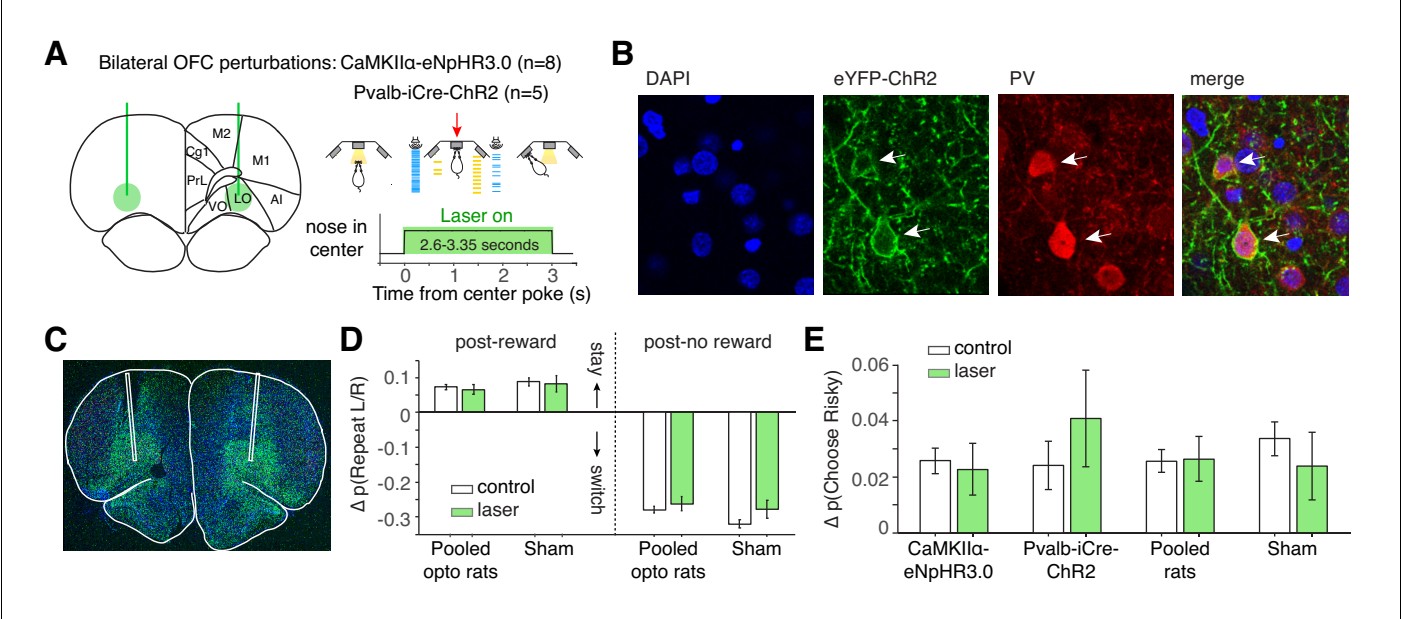

**Figure 3.** Optogenetic perturbation of lOFC during the cue period does not affect spatial or risky trial history biases. (**A**) Schematic of bilateral optogenetic perturbations. For CaMKIIα-eNpHR3.0 rats (n = 8), we used continuous illumination of a green laser for photoinhibition. For Pvalb-iCre-ChR2 rats (n = 5), a blue laser was pulsed at 20 Hz. See also *Figure 3—figure supplement 1*. While the schematic shows a 3 s trial, trial durations were variable (2.6–3.35 s); photoinhibition persisted for the duration of the cue period. (**B**) Histological section from Pvalb-iCre-ChR2 rats also stained for DAPI and parvalbumin (PV) immunoreactivity. (**C**) Virus injection in a wild type rat expressing CaMKIIα-eNpHR3.0. Location of fibers were estimated by damage at brain surface and fiber tracks. (**D**) Magnitude of spatial win-stay and lose-switch biases (difference in probability of repeating a left or right choice) on control and laser trials. Error bars are normal approximation of 95% confidence intervals (Materials and methods). (**E**) Magnitude of risky win-stay bias (difference in probability of choosing the safe option following safe or risky rewards) on control and laser trials.

DOI: https://doi.org/10.7554/eLife.49744.007

The following figure supplement is available for figure 3:

**Figure supplement 1.** Characterization of photoinhibition in Pvalb-iCre-ChR2 rats.

DOI: https://doi.org/10.7554/eLife.49744.008

CaMKIIα-eNpHR3.0 or Pvalb-iCre-ChR2 rats (*Figure 3E*; p=0.005, one-way ANOVA comparing safe choices following safe or risky rewards, pooling data across all 13 optogenetic rats).

## Disrupting lOFC during the choice report eliminated the risky win-stay bias

In contrast to during the cue period, activity at the time of the choice report, when rats exited the center poke, often reflected whether rats chose the safe or risky prospect (*Figure 4A,B*). A subset of lOFC neurons exhibited significantly different spike counts on rewarded trials when rats chose the safe compared to the risky option (*Figure 4B–E*; n = 128 units, unpaired t-test). For these neurons, discriminability peaked shortly after rats left the center poke (*Figure 4C,D*). We also observed prominent side-selectivity (n = 628 units) and strong encoding of reward receipt (n = 459) during the choice reporting period (*Figure 4F–H*). Of the neurons whose activity reflected safe/risky choice, there was no significant side bias (*Figure 4—figure supplement 1*).

Given the prominent encoding of rats' left/right choices during this period (*Figure 4F,G*), we next tested whether photoinhibition affected spatial win-stay/lose-switch biases for the left/right ports. We optogenetically perturbed lOFC during the choice reporting period (triggered when rats exited the center port) for 4 s, and then analyzed performance on subsequent trials. Inhibition during the choice reporting period was interleaved in the same sessions as inhibition during the cue period, in the animals shown in *Figure 3*. While there was a significant reduction in spatial biases, sham rats also exhibited reduced win-stay/lose-switch biases (*Figure 5B*). Photoinhibition during the choice reporting period imposed a minimum inter-trial interval (ITI) of 4 s (for laser illumination), which was

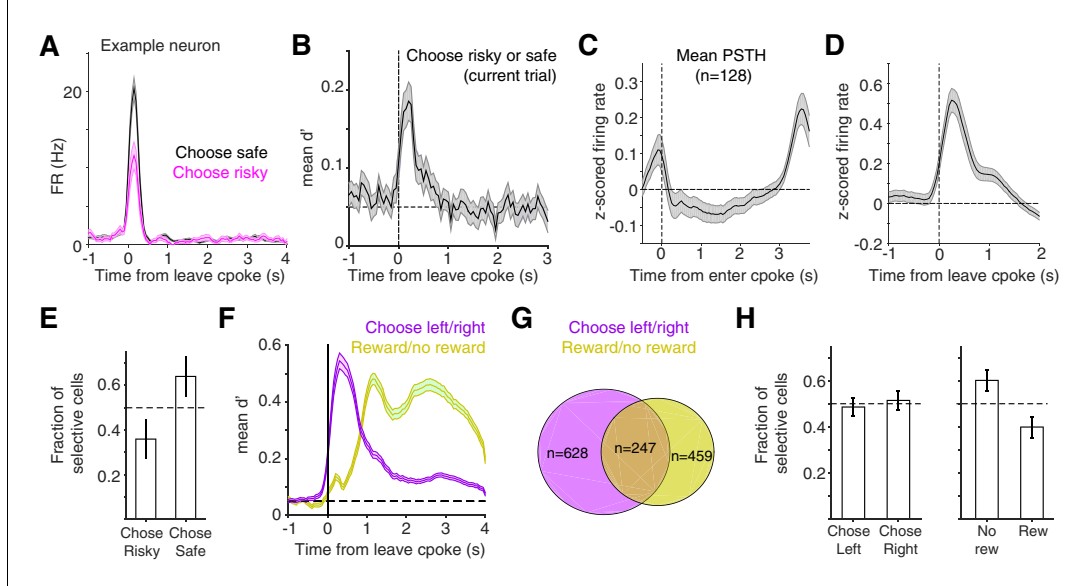

**Figure 4.** At time of choice report, lOFC neurons represent risk, reward, and left/right choice. (A) Example lOFC neuron with activity aligned to when the rat left the center poke to report his choice. This neuron's firing rate reflected whether the rat chose the risky (magenta) or safe (black) option on the current trial, analyzing rewarded trials only. (B) Mean d' across lOFC neurons with significantly different spike counts on trials with risky or safe choices. See also *Figure 4—figure supplement 1*. (C,D) Mean z-scored firing rate of neurons in panel B aligned to entering the center poke (C), or leaving it to report choice (D). (E) Fraction of neurons in panels B-D that preferred trials when rats made risky or safe choices. Higher firing rates on trials in which rats chose the safe reward could reflect encoding of decision confidence or reward expectation (*Lak et al., 2014*). (F) Mean d' reflecting whether rats chose the left/right ports, or whether rats received reward, averaged across neurons with significantly different spike counts on those trials. See also *Figure 4—figure supplement 1*. (G) Venn diagram of overlap between neurons whose activity differentiated between left/right choices and rewarded/unrewarded trials. (H) Fraction of neurons in panels F,G preferring left/right choices or rewarded/unrewarded trials.
DOI: https://doi.org/10.7554/eLife.49744.009

The following figure supplement is available for figure 4:

**Figure supplement 1.** Results do not depend on whether units are treated independently over days.
DOI: https://doi.org/10.7554/eLife.49744.010

longer than the average ITI (*Figure 1—figure supplement 1A*). Comparing the photoinhibition trials to control trials with a minimum ITI of 4 s eliminated the reduction in spatial win-stay/lose-switch biases in CaMKIIα-eNpHR3.0, Pvalb-iCre-ChR2, and sham rats (data not shown).

In contrast, optogenetic inhibition of lOFC during the choice reporting period eliminated the risky win-stay bias on the subsequent trial in rats expressing light-sensitive opsins (*Figure 5A,C*; p=0.667 CaMKIIα-eNpHR3.0; p=0.778 Pvalb-iCre-ChR2; one-way ANOVA comparing safe choices following safe or risky rewards, pooling data across rats). Laser illumination did not disrupt the risky win-stay bias in sham rats not expressing light-activated opsins (*Figure 5C*; p=0.009, one-way ANOVA comparing safe choices following safe or risky rewards, n = 3 rats). For the majority of rats (7/8 CaMKIIα-eNpHR3.0 rats and 4/5 Pvalb-iCre-ChR2 rats), there was no significant effect on choice latencies compared to control trials (Wilcoxon rank-sum test, Bonferroni correction for multiple comparisons). Moreover the behavioral effect occurred on trials *subsequent* to the optogenetic perturbations (*Figure 5A*), making it unlikely that they were due to off-target illumination of motor cortex, including overlying M2.

We wanted to determine if photoinhibition of lOFC affected other potential biases, so we used logistic regression with parameters for choice repetition for safe/risky choices, the risky win-stay bias described above, left/right choice repetition, and systematic left/right side biases (*Kagel et al., 1995*; *Padoa-Schioppa, 2013*). Photoinhibition during the choice report exclusively reduced the risky win-stay bias parameter, but not others (*Figure 5—figure supplement 1*; p=0.0063, one-way ANOVA with Bonferroni correction; Materials and methods).

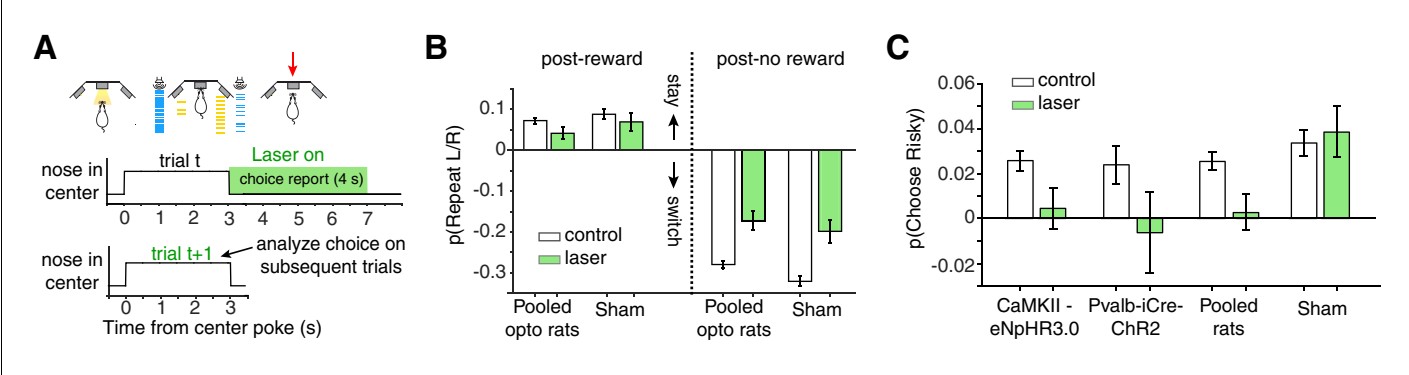

**Figure 5.** Photoinhibition of lOFC at the time of choice report selectively eliminates the risky win-stay bias. (**A**) For choice reporting period perturbations, the laser was triggered when rats left the center poke, and persisted for 4 s into the inter-trial-interval. See also *Figure 5—figure supplement 1*. (**B**) Spatial win-stay/lose-switch biases following photoinhibition during the choice reporting period; sham rats also exhibited a significant reduction in lose-switch biases, and trended towards a reduction in win-stay biases. Control data are replotted from *Figure 3D*. (**C**) Magnitude of the risky win-stay bias following choice reporting period inactivations. Control data are replotted from *Figure 3E*. Error bars are 95% confidence intervals.

DOI: https://doi.org/10.7554/eLife.49744.011

The following figure supplement is available for figure 5:

**Figure supplement 1.** Photoinhibition during the choice reporting period does not affect baseline performance, but selectively reduces the risky win–stay bias.

DOI: https://doi.org/10.7554/eLife.49744.012

The reduction of the risky win-stay bias did not reflect changes in the baseline probability of choosing safe (*Figure 5—figure supplement 1*), and was observed on a rat-by-rat basis (pooling CaMKIIα-eNpHR3.0 and Pvalb-iCre-ChR2 rats, p=0.015; paired t-test across rats). Therefore, photo-inhibition of lOFC specifically eliminated risky biases and not spatial biases. Moreover, risky biases were not sensitive to ITI duration (*Figure 5C*, sham), whereas spatial biases decreased with ITI dura-tion (*Figure 5B*, sham), further suggesting that these sequential dependencies are dissociable.

## Discussion

Across species, OFC has been implicated in myriad aspects of value-based decision-making, includ-ing representing the value of offered and chosen goods, expected outcomes, confidence, regret, and credit assignment (*Akaishi et al., 2016*; *Kepecs et al., 2008*; *Padoa-Schioppa, 2011*; *Rudebeck and Murray, 2014*; *Schoenbaum et al., 1998*; *Steiner and Redish, 2014*). Evidence from studies using reversal learning paradigms or Pavlovian instrumental transfer suggests that the OFC is critical for behavioral flexibility (*Izquierdo et al., 2004*; *Schoenbaum et al., 2003*), and recent work indicates that value representations in lOFC may drive learning rather than action selection or choice (*Miller et al., 2018*). Relatedly, a recent study showed that rats with lOFC lesions exhibited dis-rupted sensitivity to previous trial outcomes, especially when those outcomes were unexpected, analogous to probabilistic rewards in our task (*Stolyarova and Izquierdo, 2017*). Our data are con-sistent with a role for lOFC in updating rats' risk attitudes or their beliefs about the world (*Jones et al., 2012*; *McDannald et al., 2011*; *Miller et al., 2017*; *Wilson et al., 2014*). lOFC neurons reflected reward history most strongly at trial initiation, when the animal has no information yet about the prospects on the current trial (*Nogueira et al., 2017*). These dynamics are therefore dis-tinct from relative value coding observed in primate OFC, in which neurons encode the value of rewards on the current trial relative to rewards on the previous trial (*Kennerley et al., 2011*; *Padoa-Schioppa, 2009*; *Saez et al., 2017*). The temporal response profiles we observed are generally con-sistent with reports that OFC neurons fire transiently when an animal initiates reward-seeking behav-ior, here, trial initiation (*Moorman and Aston-Jones, 2014*).

We observed many 'side-selective' neurons whose activity reflected which side the rat chose. A hallmark of primate OFC is that neurons do not encode spatial location (*Grattan and Glimcher, 2014*; *Padoa-Schioppa and Cai, 2011*). This discrepancy could reflect a species difference

(*Feierstein et al., 2006*; *Kuwabara and Holy, 2019*; *Roesch et al., 2006*). Alternatively, side-selectivity could reflect encoding of the left and right prospects or 'goods' on each trial (*Padoa-Schioppa, 2011*).

This study used sensory stimuli (visual flashes, auditory clicks) to convey information about reward options to the animal. Evidence from primates suggests that some OFC neurons may be selective for particular stimulus identities used to indicate reward attributes (*Hunt et al., 2018*). An intriguing direction for future research would be to record neural activity in OFC as animals learn the associations between sensory stimuli and reward attributes, to characterize the evolution of dynamics in OFC from early to late in training.

We found that perturbation effects were uncoupled from task variables that seemed to be encoded most strongly, quantified by either fraction of neurons or discriminability, both during the cue period (reward history) and during the choice reporting period (left/right choice). This raises an intriguing question: why are these variables so strongly represented in lOFC if the animal appears to not be using those representations to guide behavior? It is possible that representations of reward history and choice may be distributed broadly enough that other brain areas can compensate for local perturbations, whereas representations used to update risk preferences may be more narrowly localized within or read out from OFC.

Are specific subcircuits within OFC responsible for the risky win-stay bias, and updating dynamic risk preferences more generally? The relatively small fraction of neurons (~9%) that reflect whether choices are risky or safe may be behaviorally relevant, perhaps occupying a privileged position in the circuit and/or projecting to a common target. Alternatively, the more substantial fraction of neurons representing whether animals received reward (~31%) may be involved in updating risk preferences, in which case their activity is read out to specifically update abstract task-specific (here, risky), but not spatial, biases. This latter hypothesis is consistent with a recent study of medial OFC (mOFC) in mice (*Namboodiri et al., 2019*). In a Pavlovian conditioning paradigm, in which a tone probabilistically predicted a sucrose reward, mice exhibited trial-by-trial updating of their reward expectation, revealed by their anticipatory licking, based on reward history. Optogenetic inactivation of mOFC neurons projecting to the ventral tegmental area (VTA) during the reward period, but not the cue period, disrupted this trial-by-trial learning (*Namboodiri et al., 2019*). While there is increasing evidence for functional differences between medial and lateral OFC in rodents, these results are consistent with the present study, and suggest that the risky win-stay bias we observed may derive from lOFC neurons projecting to the VTA.

The risky win-stay bias may reflect evolutionary pressures in dynamic foraging environments, in which sequential successful outcomes are often not independent but reflect 'clumped' resources (*Blanchard et al., 2014*; *Wilke and Barrett, 2009*). However, today it demonstrably (and often adversely) affects behavior in finance, recreational gambling, and sports (*Croson and Sundali, 2005*; *Hoffmann et al., 2010*; *Neiman and Loewenstein, 2011*). Our data show that this particular sequential bias is observable and manipulable in populations of neurons in lOFC, although lOFC's involvement may depend upon task design.

OFC has been proposed to represent the animal's location in a cognitive map of the task, which, in a reinforcement learning framework, corresponds to the current state in an abstract representation of task states and transitions between them (*Nogueira et al., 2017*; *Wilson et al., 2014*). The cognitive map hypothesis parsimoniously accounts for the results of OFC lesions in a variety of paradigms including delayed alternation, extinction, devaluation, and reversal learning, and is consistent with OFC's role in evaluative processes such as regret (*Gallagher et al., 1999*; *Izquierdo et al., 2004*; *Pickens et al., 2003*; *Steiner and Redish, 2014*; *Steiner and Redish, 2012*; *Wilson et al., 2014*). A recent study showed that the OFC may be particularly important for *learning* of action-outcome values (*Miller et al., 2018*). Our data are consistent with this hypothesis, and indicate that the coordinate space of the cognitive map in which OFC promoted learning in this task was in abstract (risky or safe), but not spatial (left or right) coordinates.

## Materials and methods

### Animal subjects

A total of 39 male rats between the ages of 6 and 24 months were used for this study. These included 35 Long-evans and 4 Sprague-Dawley rats (*Rattus norvegicus*). Of these, three rats were used for neural recordings, and 16 for optogenetic experiments, including LE-Tg (Pvalb-iCre)2Ottc rats (n = 5) made at NIDA/NIMH and obtained from the University of Missouri RRRC (transgenic line 0773). These are BAC transgenic rats expressing Cre recombinase in parvalbumin expressing neurons. Investigators were not blinded to experimental groups during data collection or analysis. Animal use procedures were approved by the Princeton University Institutional Animal Care and Use Committee and carried out in accordance with National Institutes of Health standards.

Animals were water restricted to motivate them to perform behavioral trials. They obtained water rewards during behavioral training sessions, which ranged from 1 to 5 hr per day, and an ad lib access period of up to 1 hr. Food was typically placed in the behavioral box during training, so it was available during the water access period. All rats obtained a minimum volume of water equal to 3–5% of their body mass, (30–50 mL/kg). Water consumption was monitored during the behavioral session, and if rats consumed less than the minimum requirement, additional water was offered during an ad lib period. The ad lib period terminated either when the target water volume was exceeded or after 1 hr.

### Behavior

We have previously described rats' behavior on this task in detail (*Constantinople et al., 2019*). Briefly, rats were trained in a high-throughput facility using a computerized training protocol. Rats were trained in operant training boxes with three nose ports. When an LED from the center port was illuminated, the animal could initiate a trial by poking his nose in that port; upon trial initiation the center LED turned off. While in the center port, rats were continuously presented with a train of randomly timed clicks from a left speaker and, simultaneously, a different train of clicks from a right speaker. The click trains were generated by Poisson processes with different underlying rates (*Hanks et al., 2015*); the rates conveyed the water volume baited at each side port. After a variable pre-flash interval ranging from 0 to 350 ms, rats were also presented with light flashes from the left and right side ports; the number of flashes conveyed reward probability at each port. Each flash was 20 ms in duration; flashes were presented in fixed bins, spaced every 250 ms, to avoid perceptual fusion of consecutive flashes. After a variable post-flash delay period from 0 to 500 ms, the end of the trial was cued by a go sound and the center LED turning back on. The animal was then free to choose the left or right center port, and potentially collect reward.

In this task, the rats were required to reveal their preference between safe and risky rewards. First, rats proceeded through a series of early training stages that included training the rat to center poke, gradually growing the duration of center fixation, and introducing cues representing certain rewards of each volume on one side at a time. Once they were in the final training stage they were presented with the full choice set. To determine when rats were sufficiently trained to understand the meaning of the cues in the task, we evaluated the 'efficiency' of their choices as follows. For each training session, we computed the average expected value per trial of an agent that chose randomly, and an expected value maximizer, or an agent that always chose the side with the greater expected value. We compared the expected value per trial from the rat's choices relative to these lower and upper bounds. Specifically, the efficiency was calculated as follows:

$$efficiency = 0.5 \frac{rat_{EV/trial} - rand_{EV/trial}}{EVmax_{EV/trial} - rand_{EV/trial}} + 0.5$$

The threshold for analysis was the median performance of all sessions minus 1.5 times the interquartile range of performance across the second half of all sessions. Once performance surpassed this threshold, it was typically stable across months. Occasional days with poor performance were usually due to hardware malfunctions in the rig or a change in the experiment (e.g., the first day being tethered for electrophysiological recordings). Days in which performance was below threshold were excluded from analysis.

## Psychometric curves

We measured rats' psychometric performance when choosing between the safe and risky options. For these analyses, we excluded trials where both the left and right side ports offered certain rewards. We binned the data into 11 bins of the difference in the expected value (reward x probability) of the safe minus the risky option. Psychometric plots show the probability that the subjects chose the safe option as a function of this difference (see *Figure 5—figure supplement 1*). We fit a 4-parameter sigmoid of the form:

$$p(Choose_S) = y_0 + \frac{1 - 2a}{(1 + e^{(-b(V_S - V_R - x_0))})},$$

where y0, a, b, and x0 were free parameters. Parameters were fit using a gradient-descent algorithm to minimize the mean square error between the data and the sigmoid, using the sqp algorithm in Matlab's constrained optimization function fmincon.

## Chronic electrophysiology

Tetrodes were constructed from twisted wires that were either PtIr (18 μm, California Fine Wire) or NiCr (25 μm, Sandvik). Tetrode tips were platinum- or gold-plated to reduce impedances to 100–250 kΩ at 1 kHz using a nanoZ (White Matter LLC).

Microdrive assemblies were custom-made as described previously (*Aronov and Tank, 2014*). Each drive contained eight independently movable tetrodes, plus an immobile PtIR reference electrode. Each animal was implanted over the right OFC. On the day of implantation, electrodes were lowered to ~4.1 mm DV. Animals were allowed to recover for 2–3 weeks before recording. Shuttles were lowered ~30–60 μm approximately every 2–4 days.

Data were acquired using a Neuralynx data acquisition system. Spikes were manually sorted using MClust software. Units with fewer than 1% inter-spike intervals less than 2 ms were deemed single units. All units that fired more than two spikes on half of trials were included in analysis (n = 1459/1881). To convert spikes to firing rates, spike counts were binned in 50 ms bins and smoothed using Matlab's smooth.m function.

## Discriminability, or *d'*, of OFC neurons

To measure neuronal discriminability for different task variables, such as whether the previous trial was rewarded, we computed the mean difference in the smoothed firing rate on different trial types divided by the square root of their mean variance:

$$d' = \frac{|\mu_1 - \mu_2|}{\sqrt{\frac{1}{2}(\sigma_1^2 + \sigma_2^2)}}.$$

Because we computed the absolute value of the difference in firing rates, we subtracted the mean shuffled *d'*, computed from shuffling the data 15 times. *d'* was computed in 50 ms bins, as this was the bin-width used for computing the firing rates (see above).

## Spike waveform analysis for identifying the same neurons recorded over days

The single neuron data shown in *Figures 2* and *4* treated each unit recorded on a different day/recording session as a unique unit. However, we also modified previously published methods (*Tolias et al., 2007*) to identify units recorded over days (*Figure 2—figure supplement 1*). We computed two metrics (*Tolias et al., 2007*), which we describe below, based on the spike waveform. The first metric compared how similar the shape of the waveform was across recording sessions. For each waveform on session 1 (x), we computed α to make it as close as possible to the waveform on session 2 (y):

$$\alpha(x, y) = \underset{\alpha}{\operatorname{argmin}} ||\alpha x - y||^2$$

We used Matlab's constrained minimization function fmincon.m to find α. We then computed the Euclidean distance between the scaled waveforms, $d_1$.

$$d_1(X,Y) = \sum_{i=4}^{4} \frac{||\alpha(x_i,y_i)x_i - y_i||}{||y_i||}$$

The second metric, $d_2$ quantified the difference in amplitude across the 4 channels of each tetrode.

$$d_2(X,Y) = \max_{i=4}^{4} |log(\alpha(x_i,y_i))| + \max_{i,j}^{4} |log(\alpha(x_i,y_i)) - log(\alpha(x_j,y_j))|$$

We computed these metrics for all pairs of waveforms recorded on the same tetrode on subsequent recording sessions. To compare these values to a null distribution, we computed the $d_1$ and $d_2$ metrics for units recorded from two different animals, which could not have identical waveforms. We used this null distribution to empirically determine thresholds for $d_1$ (0.8) and $d_2$ (1). Units recorded on consecutive sessions with values below these thresholds, and with significant Pearson's correlation coefficients of their mean firing rates aligned to trial start (p<0.05), were tentatively classified as identical (*Figure 2—figure supplement 1*). Putatively identical neurons were then manually examined, and those that exhibited qualitatively different PSTHs, or different mean firing rates over days were rejected and treated as separate units. 191/1459 (13%) units that met criteria for inclusion in analysis were recorded over multiple sessions. Combining data from these units over sessions did not change the results (*Figure 2—figure supplement 1*; *Figure 4—figure supplement 1*). For population analyses (TCA/PARAFAC; *Figure 2D-G*), data were not combined across recording sessions.

## Tensor components analysis/CANDECOMP/PARAFAC tensor decomposition

To fit the tensor decomposition model, we used software recently made publicly available (*Williams et al., 2018*): https://github.com/ahwillia/tensortools (copy archived at https://github.com/elifesciences-publications/tensortools).

To initially determine the dimensionality, or rank, that should be applied to each recording session, we iteratively tried different numbers of dimensions, or 'tensor components', and computed a similarity index to determine how sensitive the recovered factors were to the initialized values of the optimization procedure (*Williams et al., 2018*). The similarity index was computed on factors recovered from consecutive initializations using the score.m function in Matlab's Tensor Toolbox. The maximum number of components that yielded an average similarity index >90% was used as the number of components, or rank, for each recording session (Figure S3). Nearly all of our recording sessions were rank 1 or 2 (by this method). TCA/PARAFAC is notably different from principal components analysis (PCA) in that the first component does not necessarily explain the most variance of the data (*Williams et al., 2018*). Therefore, given that most of our data were low rank, to simplify the problem of determining which trial factors to analyze, we fit a rank one model to each recording session.

We computed the shuffle-corrected reward-triggered averages of trial factors (and no reward-triggered averages) as follows. We computed the average change in trial factors relative to the mean trial factor relative to each rewarded (and unrewarded) trial, up to seven trials in the future. We then performed a shuffle correction, shuffling the trials randomly with respect to reward history, and computed the average change in trial factors relative to rewarded and unrewarded trials (relative to the mean) for the shuffled data. We subtracted the shuffled averages from the true averages to obtain the plot in *Figure 2F*.

TCA decomposes a 3rd order data tensor $X_{n,t,k}$ (with n neurons over k trials of length t) by a sum of rank 1 factors $\sum_{r=1}^{R} w_r b_r\, a_r$ . Here, for each rank r, w is a vector of neuron factors, b is a vector of temporal (time within trial) factors, and a is a vector of across trial factors. We note that each of the factors (neuron, temporal, and trial) is a linear gain factor, multiplied by the others; therefore, meaningful units are difficult to determine (as all the factors are multiplied); their scale is a gain on the other terms.

## Acute electrophysiology

To confirm photoinhibition in Pvalb-iCre-ChR2 rats, we performed virus delivery as described below. After 6–8 weeks to allow for expression, rats (n = 2) were anesthetized with for surgery with 0.2 mL ketamine and 0.2 mL buprenorphine. Craniotomies were made over frontal orienting field (FOF; centered 2 mm anterior to the Bregma and 1.3 mm lateral to the midline), and the rat was maintained under isoflurane anesthesia.

A chemically sharpened fiber optic (50 um core, 125 um cladding) was inserted into the field of infected neurons to a depth of 1 mm. A sharp tungsten electrode (0.5 MW) mounted to a Narishige oil hydraulic micromanipulator was manually lowered into the brain. Recordings were made using a Neuralynx Cheetah system applying a bandpass filter from 300 to 6000 Hz to the voltage signal. At each site an 8 s laser illumination (473 nm, 25 mW, 20 Hz, 20 ms of pulse duration) was delivered every 20 s, 10 times. A mechanical shutter (Thor Labs optical beam shutter) was used to control the laser timing.

Spikes were automatically detected as brief (<1 ms) events that crossed a threshold of ± 5 standard deviations on the filtered voltage trace. In order to remove light artifact and possible population spikes of driven PV neurons, we removed spikes within the 20 ms of the laser pulse onset. Inhibition was defined as the mean spike rate during the 8 s laser on period/the mean spike rate during the 12 s laser off period.

## Optical fiber chemical sharpening

We used standard off the shelf FC-FC duplex fiber optic cables (#FCC2433, FiberCables.com), and as previously described (*Hanks et al., 2015*), stripped the outer plastic coating. To etch the fiber, 2–2.5 mm of the fiber tip was submerged in 48% hydrofluoric acid with mineral oil on top. Over the course of ~17 min, a motor (Narishige) slowly pulled the fiber tip out of the hydrofluoric acid, producing a long taper. The speed of the motor was then increased and maintained at a constant speed until the tip was entirely removed from the acid (usually by 13–15 min). This protocol reliably produced sharp, well-etched fibers with uniform and broad light scatter. Fibers that did not produce sufficiently broad or uniform scatter were discarded.

## Virus delivery and fiber implantation

We used methods described previously (*Hanks et al., 2015*); here we describe procedures specific to this experiment. We injected 2 µL of AAV virus (AAV5-CaMKIIα-eNpHR3.0-eYFP in wild type rats, or AAV-FLEX-rev-ChR2-tdtomato in Pvalb-iCre rats) using a Nanoject (Drummond Scientific). Six closely spaced injection tracts (typically 500 µm apart) were made in each craniotomy; each rat had bilateral craniotomies and injections, so there were 12 total injection tracts per animal. For OFC injections, in each track, 18 injections of 14.1 nL were made every 100 µm in depth starting at 3.7 mm below brain surface (3.7–5.4 mm DV). Virus was expelled at 20 nL/sec. Injections were made once every 10 s; at the final injection in a tract, the pipette was left in place for at least 2 min before removal.

Chemically sharpened fibers (50 µm core, 125 µm cladding) were implanted at 5° angles relative to the midline. Fiber tips were positioned 0.4 mm lateral from the center track at brain surface so that, when the tip was lowered to 4.6 mm DV, it was centered at the target coordinates (+3.5 AP, ± 2.5 ML). Viral constructs were allowed to develop for 6–8 weeks before behavioral experiments began.

## Optogenetic perturbation

For bilateral halorhodopsin inactivations, the laser beam from a 200 mW, 532 nm laser (OEM Laser Systems) was split into two beams of roughly equal power (~25 mW) using a beam splitter (Doric DMC_1 × 2i_VIS_FC). Laser illumination was delivered on a subset of trials by opening a shutter with a 5 V TTL. On a random subset of 15% of trials, illumination occurred during the entire trial (triggered when the rat entered the center poke until he was free to leave it), or on a random 15% of trials, illumination was triggered when the rat left the center poke, and persisted for the first 4 s of the inter-trial interval, resulting in transient silencing of cortical dynamics. In Pvalb-iCre-ChR2 rats, we used a 473 nm laser. Laser pulses (10 ms pulse width) were delivered at 20 Hz.

The rats generally chose risky on a minority of trials, and by definition, the fraction of those that were rewarded is a smaller subset. Moreover, rewarded trials were more frequent than non-rewarded trials. To overcome these challenges, we collected data from many sessions. The median number of sessions in which rats experienced photoinhibition was 27 (range: 21–70, mean: 37). We did not observe differences in the optogenetic result when comparing the first or second half of all sessions (data not shown).

## Normal approximation of 95% binomial confidence intervals

To compute error bars on the choice probabilities for the risky and spatial win-stay biases (*Figures 3D,E* and *5B,C*), we computed the 95% confidence intervals using the test statistic for the chi-square distribution (*Corder and Foreman, 2014*) as follows:

$$CI_{95} = z\sqrt{\frac{p_{psr}(1-p_{psr})}{n_{psr}} + \frac{p_{prr}(1-p_{prr})}{n_{prr}}}$$

where $p_{psr}$ is the probability of choosing safe following a safe reward ('post-safe reward') and $p_{prr}$ is the probability of choosing safe 'post-risky reward', n is the number of observations for each condition (post-safe reward and post-risky reward), and z the z-score for 95% confidence intervals from a normal distribution.

## Logistic regression model of choice biases

To evaluate the contribution of different potential choice biases to behavior, we implemented a logistic regression model, in which we parameterized the rats' probability of choosing right as follows:

$$p(ChooseR) = 1/(1 + exp(\Delta EV + rs_{hyst} + risky_{win-stay} + lr_{hyst} + lr_{bias})),$$

where $\Delta EV$ is the right minus left expected value (reward x probability) on each trial, $rs_{hyst}$ captures risky/safe hysteresis (i.e., if the rat just chose safe, the likelihood he will choose safe on the next trial), $risky_{win-stay}$ parameterizes increased willingness to choose the gamble conditioned on a risky win, $lr_{hyst}$ parameterizes the probability of repeating left/right choices, and $lr_{bias}$ parameterizes overall side biases for the left and right port. The only parameter that was significantly changed (and in fact reduced) by photoinhibition during the choice report was the $risky_{win-stay}$ parameter (p=0.0063, one-way ANOVA comparing parameters fit to control and opto conditions across rats, Bonferroni correction for multiple comparisons).

## Acknowledgements

We thank Paul Glimcher, Kenway Louie, Mike Long, David Schneider, Ben Scott, Emily Dennis, Mikio Aoi, Matthew Lovett-Barron, Cristina Domnisoru, Alejandro Ramirez and members of the Brody lab for helpful discussions and comments on the manuscript. We thank Alex Williams for feedback on the manuscript and for providing guidance and software for implementing TCA/PARAFAC tensor decomposition analysis. We thank Claudia Farb, Adam Carter, and Mike Hawken for reagents and assistance with histology and confocal imaging. We thank J Teran, K Osorio, L Teachen, and A Sirko for animal training. This work was funded in part by a K99/R00 award from NIMH (MH111926, to CMC).

## Additional information

### Funding

| Funder | Grant reference number | Author |
| --- | --- | --- |
| National Institutes of Health | K99/R00 MH111926-03 | Christine M Constantinople |

The funders had no role in study design, data collection and interpretation, or the decision to submit the work for publication.

## Author contributions
Christine M Constantinople, Conceptualization, Resources, Data curation, Software, Formal analysis, Funding acquisition, Validation, Investigation, Visualization, Methodology, Writing—original draft, Project administration, Writing—review and editing; Alex T Piet, Writing—review and editing, Analysis and interpretation of data; Peter Bibawi, Investigation, Visualization; Athena Akrami, Data curation, Investigation, Visualization; Charles Kopec, Investigation, Writing—review and editing; Carlos D Brody, Supervision, Writing—review and editing, Interpretation of data

## Author ORCIDs
Christine M Constantinople ⓘ https://orcid.org/0000-0003-4435-4460
Alex T Piet ⓘ https://orcid.org/0000-0002-6529-1414
Carlos D Brody ⓘ http://orcid.org/0000-0002-4201-561X

## Ethics
Animal experimentation: This study was performed in accordance with the National Institutes of Health standards. Animal use procedures were approved by the Princeton University Institutional Animal Care and Use Committee (protocol #1853).

## Decision letter and Author response
Decision letter https://doi.org/10.7554/eLife.49744.015
Author response https://doi.org/10.7554/eLife.49744.016

# Additional files

## Supplementary files
• Transparent reporting form DOI: https://doi.org/10.7554/eLife.49744.013

## Data availability
Data generated during this study are included in the manuscript and supporting files.

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
