## [Decision Letter]

Thank you for submitting your article "Orbitofrontal cortex promotes trial-by-trial learning of risky, but not spatial, biases" for consideration by *eLife*. Your article has been reviewed by two peer reviewers, including Emilio Salinas as the Reviewing Editor and Reviewer #1, and the evaluation has been overseen by Floris de Lange as the Senior Editor. The following individual involved in review of your submission has agreed to reveal their identity: Alicia Izquierdo (Reviewer #2).

The reviewers have discussed the reviews with one another and the Reviewing Editor has drafted this decision to help you prepare a revised submission.

Summary:

Constantinople et al. study sequential biases, or the influence of past experience, on decision making. They point out a shortcoming in many similar investigations: that choice options are fixed in space and therefore make it impossible to study reward history effects independent of spatial information. Here, rats learn to combine reward probability with reward amount in a novel, two-alternative forced-choice task, and exhibit two notable effects: (1) the classic "win-stay" and "lose-switch" biases in favor of previously rewarded or against previously unrewarded locations, and (2) a preference toward the risky (low-probability) option following a previously rewarded, risky choice. In other words, when the rat gambles and wins, it is more likely to gamble again. Electrophysiology and optogenetics reveal that OFC activity encodes the history of previous rewards, the preference for risk, and the left/right choices. And somewhat surprisingly, when OFC is inactivated during the choice period of trial j, the preference for risk in trial j+1 essentially disappears, whereas the spatial bias (win-stay/lose-switch) is unchanged. This is an interesting study that provides nuanced new information about OFC contributions to decision making.

Essential revisions:

1) Revise some of the Introduction to include a narrower set of studies as the basis for these experiments. For example, currently there is cited evidence on mOFC (Bradfield et al., Neuron, 2015) in unobservable outcome encoding, yet there is additional (and growing) evidence for functional dissociations of rat mOFC from lOFC (Izquierdo, 2017) as is presently studied here; see most recent relevant example of this by Hervig et al., 2019. Do authors attribute this specific function to lOFC or is it also possibly mediated by other subregions of OFC? I also found the conceptual connections of the present work on the influence of reward history/statistics with the economic decision making literature in the Introduction rather loose. What about this task makes it an economic decision making task?

2) During the cue period there was multimodal signaling of magnitude (click rate, auditory) and probability (light flashes, visual). Authors may wish to consider how these complex, compound stimuli experiences may change the involvement of OFC over the entire session and across sessions, particularly given that these are longitudinal experiments. For example, are these different reward attributes represented in an orthogonal way in OFC, and do they change over time (from early vs. late learning)? There is evidence of this in nonhuman primate PFC (Hunt et al., 2018) and the contrast could be interesting. Relatedly, authors could discuss if/how other regions (e.g. ACC, M2) in rat may multiplex these reward attributes and contribute to choice and action selection. Or is the authors' stance that OFC is doing this? Please clarify.

Figures 3JD,E, 4J and L5B,C3) The viral expression photomicrograph looks well-focused mostly on lateral orbital but I also see expression in ventral orbital, and perhaps even in M2. It may be more thorough to show a multi-coronal section reconstruction at different AP levels of this viral spread and correlate positioning to behavioral effects.

4) The bias for risk is quite small; if I understood correctly, it amounts to an average change in choice probability of 0.02, which is about 10 times smaller than the average lose-switch bias (Figure 1F, H). This is consistent with the much weaker coding of risk bias versus spatial bias (Figure 4B, F). This is fine, it is what it is, but the authors should mention this difference a bit more explicitly. Initially, it is surprising that the inactivation removes the risk bias but leaves intact the spatial bias, given how important the latter seems. The explanation offered in the Discussion ("We found that perturbation effects…") is perfectly reasonable, but the size difference could be pointed out to emphasize that there is still a bit of a mystery to be solved there, i.e., why is the spatial bias so strongly represented in OFC, if its primary function is not to enforce it?

Related to this, the units of the risky bias change. In Figure 1F, G, it is just the actual change in probability, whereas later on (Figures 3E, 5C) it is expressed as a percentage, but a percentage of what? Wouldn't it be clearer if the numbers indicated the raw difference in probability, as in Figure 1F, G, even if they were small? By the way, the caption of Figure 3E currently describes the units of the risk bias as a "difference in probability," not as a percentage.

5) If I understand correctly, the trial factor in the TCA decomposition represents the gain of each trial. Doesn't this factor, and thus the result in Figure 2G, depend on the number of recorded neurons? It would be useful to say something about this in the Materials and methods, because the limited number of recorded neurons might underestimate the correlation between trial factor and reward history. Also, the significance of the correlations might depend on the number of recorded neurons, no?

6) Results subsection “Disrupting lOFC during the choice report eliminated the risky win-stay bias”, first paragraph: "and the majority of those units were not selective for left/right choice." This statement seems incorrect. Figure 4—figure supplement 1 shows that 54 neurons were not selective for side, whereas 47+27=74 *were* selective.

7) Another point concerns the OFC and timing. There are now at least 2 studies showing that rodent OFC may be important in reinforcement timing (Bakhurin et al., J Neurosci, 2017; Stolyarova and Izquierdo, 2017). Specifically, the latter reported evidence that OFC lesions reduce the animal's ability to represent a detailed delay distribution of reward delivery times, acquired over longitudinal experience (as in the present study). Perhaps I missed this, but have authors considered if/how the time window of the cue period impacts strategy/performance during the reporting period? It would be good to know that there are no differences in choice reporting that can be attributed to a failure in representing short (2.6 s) vs. long (3.4 s) cue periods. Even if this were the case, it could help address the possibility that OFC is recruited to representing the variance of a temporal quantity. Relatedly, I'm not clear on why 2.6-3.35 sec range was chosen, and 4 sec range for photoinhibition imposed, can authors provide a rationale or clarify?

---

## [Author Response]

Essential revisions:1) Revise some of the Introduction to include a narrower set of studies as the basis for these experiments. For example, currently there is cited evidence on mOFC (Bradfield et al., Neuron, 2015) in unobservable outcome encoding, yet there is additional (and growing) evidence for functional dissociations of rat mOFC from lOFC (Izquierdo, 2017) as is presently studied here; see most recent relevant example of this by Hervig et al., 2019. Do authors attribute this specific function to lOFC or is it also possibly mediated by other subregions of OFC? I also found the conceptual connections of the present work on the influence of reward history/statistics with the economic decision making literature in the Introduction rather loose. What about this task makes it an economic decision making task?

We thank the reviewer for this comment. We have rewritten the Introduction, including a discussion of the different subdivisions of OFC. We have removed the Bradfield reference, and cited the papers the reviewer suggested. We have also removed the section of the Introduction that discussed economic decision-making w.r.t. OFC. We have cut and pasted some of the new text from the revised Introduction below:

“The OFC is not a monolithic structure: in rats, subdivisions (e.g., ventral orbital area, lateral orbital area, agranular insula) are characterized by distinct efferent and afferent projections and, presumably as a consequence, there is growing evidence that these subdivisions make distinct functional contributions to behavior (Dalton et al., 2016; Groeneweggen, 1988; Hervig et al., 2019; Izquierdo, 2017; Murphy and Deutch, 2018; Ray and Price, 1992). Based on connectivity, the rat lOFC (including lateral orbital and agranular insular areas) is thought to be homologous to the central-lateral OFC of monkeys (Heilbronner et al., 2016; Izquierdo, 2017; Stalnaker et al., 2015), although differences have been observed in these areas across species, for example in neural dynamics following reversal learning (Morrison et al., 2011; Schoenbaum et al., 1999).”

2) During the cue period there was multimodal signaling of magnitude (click rate, auditory) and probability (light flashes, visual). Authors may wish to consider how these complex, compound stimuli experiences may change the involvement of OFC over the entire session and across sessions, particularly given that these are longitudinal experiments. For example, are these different reward attributes represented in an orthogonal way in OFC, and do they change over time (from early vs. late learning)? There is evidence of this in nonhuman primate PFC (Hunt et al., 2018) and the contrast could be interesting. Relatedly, authors could discuss if/how other regions (e.g. ACC, M2) in rat may multiplex these reward attributes and contribute to choice and action selection. Or is the authors' stance that OFC is doing this? Please clarify.

We thank the reviewer for raising this thoughtful question. We are particularly interested in whether the different reward attributes in this task are represented in an orthogonal way in OFC. Due to the large choice set (each panel in Figure 1E corresponds to a unique trial type) and limited data, however, addressing this question with our dataset is non-trivial. We are currently developing more sophisticated analytical tools to reveal how neurons in OFC represent these different reward attributes, but this is an ongoing research effort that we view as being outside the scope of the present manuscript. We hope to have a thorough answer for the reviewer in a future manuscript.

We have modified the Discussion to cite recent evidence suggesting that the OFC may be important for learning of action values, but does not participate in action selection or choice per se. At the reviewer’s suggestion we now reference Hunt et al. (2018). We have included the following text in the revised Discussion:

**“**This study used sensory stimuli (visual flashes, auditory clicks) to convey information about reward options to the animal. […] An intriguing direction for future research would be to record neural activity in OFC as animals learn the associations between sensory stimuli and reward attributes, to characterize the evolution of dynamics in OFC from early to late in training.”

3) The viral expression photomicrograph looks well-focused mostly on lateral orbital but I also see expression in ventral orbital, and perhaps even in M2. It may be more thorough to show a multi-coronal section reconstruction at different AP levels of this viral spread and correlate positioning to behavioral effects.

We thank the reviewer for this comment. We intentionally made large virus injections, because we reasoned that the spread of light from the fiber tip (which our control experiments have shown to be approximately 1mm) would determine and limit the inactivated area. The photomicrograph shown in the main and supplementary figure was selected because it was possible to see the track of the fiber optic in this image (Figure 3—figure supplement 1E). Unfortunately, this is not representative: because we used relatively small optic fibers (50μm core, compared to 200μm or 400μm core used by most labs), in most cases, the fibers did not produce visible damage in the tissue.

While there may have been some viral spread to M2, the optogenetic effect we observed occurred on trials *subsequent* to optogenetic perturbation. Moreover, in 11/13 rats, we did not observe any significant change in choice latencies during photoinhibition (i.e., the time from leaving the center poke to poking in the side port, which occurred when the laser was on). Together, these data suggest that the behavioral effect we observed was not due to off-target inhibition of M2. This reasoning is included in the Results section of the manuscript.

We have now included an additional photomicrograph (Figure 2—figure supplement 1D) showing the location of electrolytic lesions made by tetrodes in one of our rats used for electrophysiology. We hope this provides additional information and confidence in our ability to target LO/AI.

4) The bias for risk is quite small; if I understood correctly, it amounts to an average change in choice probability of 0.02, which is about 10 times smaller than the average lose-switch bias (Figure 1F, H). This is consistent with the much weaker coding of risk bias versus spatial bias (Figure 4B, F). This is fine, it is what it is, but the authors should mention this difference a bit more explicitly. Initially, it is surprising that the inactivation removes the risk bias but leaves intact the spatial bias, given how important the latter seems. The explanation offered in the Discussion ("We found that perturbation effects…") is perfectly reasonable, but the size difference could be pointed out to emphasize that there is still a bit of a mystery to be solved there, i.e., why is the spatial bias so strongly represented in OFC, if its primary function is not to enforce it?

We thank the reviewer for raising this issue. We have attempted to more explicitly address the unequal magnitude of the spatial and risky biases, by including the following text in the Results section:

“It is notable, however, that the magnitude of rats’ spatial win-stay/lose-switch biases was much larger than the magnitude of their risky win-stay biases (Figure 1F, H).”

And in the Discussion paragraph:

“This raises an intriguing question: why are these variables so strongly represented in lOFC if the animal appears to not be using those representations to guide behavior?”

Related to this, the units of the risky bias change. In Figure 1F, G, it is just the actual change in probability, whereas later on (Figures 3E, 5C) it is expressed as a percentage, but a percentage of what? Wouldn't it be clearer if the numbers indicated the raw difference in probability, as in Figure 1F, G, even if they were small? By the way, the caption of Figure 3E currently describes the units of the risk bias as a "difference in probability," not as a percentage.

We thank the reviewer for this comment, we have changed the figures so that they are all in units of Δprobability.

5) If I understand correctly, the trial factor in the TCA decomposition represents the gain of each trial. Doesn't this factor, and thus the result in Figure 2G, depend on the number of recorded neurons? It would be useful to say something about this in the Materials and methods, because the limited number of recorded neurons might underestimate the correlation between trial factor and reward history. Also, the significance of the correlations might depend on the number of recorded neurons, no?

We thank the reviewer for raising important concerns about our methods. The reviewer is correct that the trial factor in the TCA decomposition is a gain of each trial. This trial factor and the temporal factor depend on the number of recorded neurons only in the sense that if neurons exhibit heterogeneous responses during behavior, recording more neurons could sample different dynamics. Our estimate of the correlation between trial factor and reward history is the noisiest for sessions with a small number of neurons, as single neuron dynamics have a bigger effect. However, we find no systematic bias in our estimate of the correlation with reward history and the number of neurons in each session (Figure 2—figure supplement 2I). We have added the following text to the manuscript:

“We note that there was no systematic relationship between the correlation of rats’ reward history and the trial factors, and the number of simultaneously recorded neurons in each session (Figure 2—figure supplement 2H, I).”

6) Results subsection “Disrupting lOFC during the choice report eliminated the risky win-stay bias”, first paragraph: "and the majority of those units were not selective for left/right choice." This statement seems incorrect. Figure 4—figure supplement 1 shows that 54 neurons were not selective for side, whereas 47+27=74 were selective.

We thank the reviewer for catching this incorrect statement. We have removed that text from the manuscript.

7) Another point concerns the OFC and timing. There are now at least 2 studies showing that rodent OFC may be important in reinforcement timing (Bakhurin et al., J Neurosci, 2017; Stolyarova and Izquierdo, 2017). Specifically, the latter reported evidence that OFC lesions reduce the animal's ability to represent a detailed delay distribution of reward delivery times, acquired over longitudinal experience (as in the present study). Perhaps I missed this, but have authors considered if/how the time window of the cue period impacts strategy/performance during the reporting period? It would be good to know that there are no differences in choice reporting that can be attributed to a failure in representing short (2.6 s) vs. long (3.4 s) cue periods. Even if this were the case, it could help address the possibility that OFC is recruited to representing the variance of a temporal quantity. Relatedly, I'm not clear on why 2.6-3.35 sec range was chosen, and 4 sec range for photoinhibition imposed, can authors provide a rationale or clarify?

We thank the reviewer for raising this important issue. Trial durations reflect aspects of the sensory stimuli we used. Specifically, on each trial, one port offered a certain reward, which in most versions of the task corresponded to presentation of 10 flashes from that side. Flashes occurred in 250ms bins, to avoid a phenomenon known as “flicker fusion,” or the perception of rapid intermittent flashes as continuous illumination. In rats, flicker fusion is thought to occur when subsequent flashes are presented within 50ms or less, based on evoked-related potential measurements from the scalp. 250ms was chosen because it is safely outside that window; we used similar inter-flash intervals in our previous studies of perceptual decision-making (Scott et al., 2015). 250ms windows x 10 flashes imposes a minimum trial duration of 2.5 seconds. There was also a pre-flash and post-flash delay period, both of which were variable. We introduced this variability so that the various task events, including onset of the presentation of the flashes and clicks, were not perfectly correlated and could be independently related to behavior/neural responses. Requiring rats to maintain fixation by keeping their nose in a central port becomes challenging with longer durations; therefore, we did not extend the range to be much greater than the duration required to account for 10 flashes and these two delay periods.

We happened to perform some of the electrophysiological recordings first, and noticed that most of the encoding of left/right choice and reward in the inter-trial interval had ceased by 4 seconds after rats left the center port (Figure 4F). This observation motivated our choice of photoinhibiting for 4 seconds, triggered when rats left the center port.

To address the reviewer’s comment, we separately analyzed our optogenetic data for trials with short (<3 seconds) and longer (>3 seconds) durations. We found that photoinhibition during the choice report eliminated the risky win-stay bias regardless of trial duration (p=0.07, following short trials, p = 0.10 for long trials, one-way ANOVA comparing safe choices following safe or risky rewards). We note that, although the trial durations were variable, they spanned a narrow range compared to the reward delay durations (mean of 5-20 seconds) in Stolyarova and Izquierdo (2017). Bakhurin et al. (2017) used shorter reward delays (2.5 s), but in a Pavolivian conditioning paradigm, in which a conditioned stimulus predicted a reward 2.5 s later. In our task, the rats tended to choose the side offering the greater expected value, and so were using the sensory stimuli on each trial to inform that decision. Therefore, when they entered the center poke, they presumably were accumulating sensory evidence to obtain information about reward options on each side; not merely estimating elapsed time as in Bakhurin et al. Finally, once the rat was free to leave the center poke, there was no variability in reward timing imposed by the experimenter: as soon as the rat entered the side poke, he received a water reward 100ms later (or an immediate auditory cue that indicated reward omission).

Therefore, we feel that our task design and existing dataset are not well-suited to address the potential role for the OFC in representing the variance of a temporal quantity (although we do not view our data as being inconsistent with this hypothesis). This is an interesting topic for future research on the OFC, especially given the intriguing results in Bakhurin et al. (2017) and Stolyarova and Izquierdo (2017).